# A Temperature-Insensitive Resonant Pressure Micro Sensor Based on Silicon-on-Glass Vacuum Packaging

**DOI:** 10.3390/s19183866

**Published:** 2019-09-07

**Authors:** Pengcheng Yan, Yulan Lu, Chao Xiang, Junbo Wang, Deyong Chen, Jian Chen

**Affiliations:** 1State Key Laboratory of Transducer Technology, Institute of Electronics, Chinese Academy of Sciences, Beijing 100190, China (P.Y.) (Y.L.) (C.X) (J.C.); 2School of Electronic, Electrical and Communication Engineering, University of Chinese Academy of Sciences, Beijing 100049, China

**Keywords:** resonant pressure micro sensor, temperature-insensitive, silicon-on-glass cap, MEMS

## Abstract

This paper presents a temperature-insensitive resonant pressure sensor, which is mainly composed of a silicon-on-insulator (SOI) wafer for pressure measurements and a silicon-on-glass (SOG) cap for vacuum packaging. The variations of pressure under measurement bend the pressure sensitive diaphragm and regulate the intrinsic frequencies of the resonators in the device layer. While, variations of temperature cannot significantly change the intrinsic frequencies of the resonators, due to the SOG cap to offset generated thermal stress. Numerical simulations, based on finite element analysis, were conducted to calculate the residual thermal stress and optimize the sensing structures. Experimental results show that the Q-factors of the resonators are higher than 16,000, with a differential pressure sensitivity of 11.89 Hz/kPa, a nonlinearity of 0.01% F.S and a low fitting error of 0.01% F.S with the pressure varying from 100 kPa to 1000 kPa. In particular, a temperature sensitivity of ~1 Hz/°C was obtained in the range of −45 °C to 65 °C, which is one order of magnitude lower than the previously reported counterparts.

## 1. Introduction

High-precision silicon-based pressure sensors are increasingly used in the fields of automotive industries and medical electronics and aerospace [1,2,3]. Compared to other kinds of pressure sensors, such as capacitive pressure sensors, piezoelectric pressure sensors, and piezoresistive pressure sensors, resonant pressure sensors are featured with high accuracies, high resolutions, quasi-digital outputs and long-term stabilities [4]. Resonant pressure sensors are functioned with intrinsic frequency shifts of resonators, due to changes in axial stresses. However, the axial stresses are sensitive to temperature variances for the mismatches of the coefficients of thermal expansion (CTE) between silicon-based sensing elements and packaging materials [5]. Minimizing intrinsic frequency shifts, caused by temperature disturbances, has been a primary research focus for many years [6,7].

Various approaches focusing on packaging materials [8], assembling styles [9], and stress isolation structures [10,11] were proposed to reduce temperature influences. In general, packaging materials with CTE close of that of silicon, such as Pyrex7740 and BOROFLOAT®33, were used to seal the dies [8]. Meanwhile, the fragile silicon resonators were generally housed in vacuum chambers rather than over-molded package, which can protect resonators from hostile environments [9]. Moreover, several designs of stress isolation structures were proposed to address the issue of temperature shifts. For instance, an anisotropically etched intermediate layer [10] and a metal interposer structure [11] were designed to relax contacting stresses. Previously, we presented a stress isolation structure for resonant pressure sensors, which was realized by mounting the sensor die to the metal substrate, which was fixed at a corner of the sensor through stacks of small silicon dies with silicone adhesive [12]. However, key parameters, such as temperature sensitivity, temperature hysteresis, long-term stability, and offset were still affected by the residual stresses which resulted from anodic bonding and packaging processes.

In this paper, a new type of vacuum packaging, where a silicon-on-glass (SOG) cap was used for reducing the temperature sensitivity of resonant pressure sensors, was proposed. The packaging process bonded a silicon wafer to the glass cap so that the residual thermal stress between the silicon-on-insulator (SOI) wafer and the glass cap could be balanced. The fabrication processes of the pressure sensor, including SOI wafer fabrications and SOG cap fabrications, were thoroughly studied. The effects of the SOG cap on the temperature sensitivity of the sensors were also characterized.

## 2. Methodology 

### 2.1. Working Principle: 

The proposed resonant pressure sensor is shown in Figure 1a, which mainly consists of a pressure sensitive diaphragm (length: 5000 μm, width: 5000 μm) in the handle layer, a pair of H-shaped doubly-clamped resonators (length: 1400 μm, width: 20 μm, thickness: 40 μm) in the device layer, and an SOG cap with a cavity (length: 5000 μm, width: 5000 μm, thickness: 100 μm) in the glass layer. The silicon-glass bonded wafer could reduce the thermal mismatches between the SOI die and the glass wafer during anodic bonding [13,14,15]. The resonators are located in the relevant middle and the edge of the square diaphragm, which are named “central beam”, and “side beam”, respectively. In operation, with the deflections of the pressure-sensitive diaphragm, caused by pressure under measurement, the direction of the intrinsic resonant frequency shifts of the two resonators is opposite each other, as shown in Figure 1c. Thus, the differential outputs can be used to minimize the temperature disturbance and amplify the sensitivities of pressure measurement. Electromagnetic excitation/electromagnetic detection was used in this paper to measure the resonant frequencies of the resonators, as shown in Figure 1d. The detailed detection principle is that the resonator, which is perpendicular to a static magnetic field and carries an AC current, experiences a Lorentz force, drives the resonant beam into vibration, and the resulting vibration produces a magnetic induction voltage, that we can use to obtain the frequency of the resonator by detecting the voltage signal.

### 2.2. Thermomechanical Mechanism and Simulations: 

Due to the mismatches of CTE, between silicon-based sensing elements and packaging materials, temperature changes will generate residual thermal stress at the bonding interface. Figure 2a shows that the glass contracted faster than silicon while the temperature dropped from bonding temperature to ambient working temperature. According to thermomechanical mechanisms [16], the residual thermal stress at the interface between silicon and glass should be,
(1)$$\sigma =\frac{E_{2}\left(\alpha_{2}-\alpha_{1}\right)\Delta T}{1+\frac{E_{2}t_{2}}{E_{1}t_{1}}}$$ where $E_{1},E_{2}$ are Young’s modulus, $t_{1},t_{2}$ are the thickness and $\alpha_{1},\alpha_{2}$ are coefficients of thermal expansion of glass and silicon, respectively. A silicon wafer without features was bonded to the glass substrate to offset the thermal stress. Assuming that CTE of the bi-layer is $\alpha_{3}$, it is closer to $\alpha_{2}$ than $\alpha_{1}$. In addition, the glass layer was thinned using chemical mechanical planarization to make $E_{2}t_{2}\gg E_{1}t_{1}$ so that the residual thermal stress could be further reduced, as shown in Figure 2b.

Numerical simulations, based on finite element analysis (FEA), were conducted to reveal the relationships between the residual thermal stresses and the intrinsic frequency shifts of the resonators, in order to optimize the sensor structures. More specifically, thermal stresses, based on a multi-mode of steady-state thermal analysis and static structural analysis, were first calculated. The materials used in simulations are listed in Table 1. To reduce the master degrees of freedom of the entire model, two element sizes were used in these simulations. A meshing size of 100 μm was used to mesh the entire body, except the resonant beams, which were meshed in an element size of 10 μm. Then, the thermal stresses were calculated as the temperature dropped from bonding temperature (350 °C) to ambient working temperature (−55 °C~85 °C). Finally, a modal analysis was conducted to obtain the intrinsic resonant frequencies by using the thermal stresses as loads. 

To analyze the side effects of temperature disturbances, three types of packaging methods were modelled and compared, where the intrinsic frequency shifts of the two resonators as functions of temperature variances, were calculated. Figure 3a–c represent the residual thermal stress distributions of the SOI wafers after anodic bonding. The equivalent stresses in the SOI wafer bonded with the pure glass cap (500 μm in thickness), the SOG cap I (300 μm-thick silicon and 500 μm-thick glass) and the SOG cap II (300 μm-thick silicon and 50 μm-thick glass) were noticed to decrease successively, which indicates that the thermal stress could be reduced by adjusting the thickness of the glass substrate. The resonant frequencies of the resonators were calculated at a pressure of 100 kPa and a temperature of 22 °C (see Figure 3d), where the devices capped with the SOG cap II exhibit minimum frequency shift among the three types of devices. Furthermore, the temperature sensitivities of the three types of devices were quantified as 18.02 Hz/°C, 8.75 Hz/°C, and 0.42 Hz/°C, respectively, under the pressure of 100 kPa (see Figure 3e). Thus, an optimized thickness of 50 μm of the glass substrate was chosen to seal the die.

## 3. Fabrication

The fabrication processes of the sensor, include the fabrications of the SOI wafer and the SOG cap. A 4 inch SOI wafer ((100) plane, <100>oriented, p-type) with a device layer of 40 μm, a buried SiO2 layer of 2 μm and a handle layer of 300 μm was utilized. Three photolithographic and deep reactive iron etch (DRIE) steps were used to form the resonators, the pressure-sensitive diaphragm and the through silicon vias (TSVs). Firstly, the SOI wafer was cleaned by a standard wafer cleaning process. Then, a 140 μm-thick pressure sensitive diaphragm and 300 μm-deep TSV holes were etched on the handle layer by DRIE (see Figure 4b). To realize the uniform thickness of the diaphragm, a patterned ZnO film of 1000 Å and 5.4 μm-thick positive photoresist (AZ4620) were used as aligned masks in DRIE [17]. Afterwards, the resonators in the device layer were fabricated by DRIE with a patterned photoresist mask and the sacrificial layer release-etch (see Figure 4c,d). 

For the SOG cap, the process started with anodic bonding between a 300 μm-thick Si wafer and a 300 μm-thick glass wafer, under a voltage of 800 V and a surrounding temperature of 350°C (see Figure 4e). Then the chemical mechanical planarization was used to thin the glass layer to the final thickness of 50 μm (see Figure 4f). Then the compound substrate was thoroughly cleaned to remove organic residues and other particulates. The getter cavity was defined by etching about 35 μm-depth cavity, where a sputtered Cr/Au film of 300 Å/3000 Å and 5.4 μm-thick photoresist functioned as compound masks (see Figure 4g). The depth of the cavity defined the distance between the resonators and the cap, which affected the temperature sensitivity as well. After stripping the mask for the glass cavity, a Ti/Au film of 5000 Å/300 Å was sputtered inside the cavity as the getter layer, which was used to absorb the gases generated during anodic bonding, ensuring that the resonators work at a high vacuum. (Figure 4h). The resonators were sealed in the vacuum chamber by bonding the SOG wafer with the fabricated SOI wafer at a low pressure of about 0.1 Pa under a voltage of 250 V and a temperature of 350 °C (see Figure 4i). Finally, a Cr/Au film was sputtered into the TSVs as electrodes. 

The proposed resonant pressure micro sensor was successfully made by bulk-silicon micromachining technology. As shown in Figure 5a, the etching depth and the undercut of the fabricated SOG wafer were measured by scanning electron microscopy (SEM), where a 50.8 μm-deep cavity and an undercut of 79.4 μm were found. Figure 5b,c shows the cross section and the top view of a suspended resonator, respectively, where an undercut of 16.6 μm was found on the anchor. The lateral etching in the oxide film was so small, that its effects on the rest of the structure could be ignored. Figure 5d shows the image of a prototype sensor with a dimension of 10 mm × 10 mm. Figure 5e shows the package prototypes of a sensor unit after wafer dicing.

## 4. Characterizations

An E5061B Network Analyzer (Agilent, USA) was utilized to obtain the frequency response of the fabricated sensor ship in an open-loop scanning manner (see Figure 6a). The resonant frequencies of the resonators were quantified as ~67,736 Hz with the phase shift of ~180° and Q-factors of 16,589, which indicated the reliability of the triple-stack anodic bonding of SOI-SOG, as shown in Figure 6b. In addition, multiple cycles of open-loop testing were conducted to obtain the shifts of the resonant frequencies (see Figure 6c,d). By comparing the pressure sensors with glass or SOG caps, it was found that the resonant pressure sensor with the SOG caps produced less shifts of the resonant frequencies when the testing cycles were increased from one to six, which demonstrates that the proposed resonant pressure sensors are more stable than the pressure sensors with glass caps. 

Furthermore, a closed-loop self-oscillating system was developed to characterize the performances of the proposed sensors, as shown in Figure 7a. The system mainly includes, an amplifier circuit, multiple filter circuit, drive buffer circuit, and automatic gain control (AGC) circuit. The induced voltage of the resonators was amplified by an instrument amplifier, and then the signal was sent to the driving end of the resonator after being attenuated and buffered. An automatic gain control circuit, including a band-pass filter, a comparator, a rectifier and a junction field effect transistor was developed to maintain the stable vibrations of the resonators. The intrinsic frequencies of the resonators were detected under various pressure and temperature, controlled by a pressure calibrator (PPC4, Fluck Co., Phoenix, AZ, USA) and a temperature and humidity chamber (SH-241, ESPEC Co., Osaka, Japan). 

In this study, the devices were characterized with a pressure range from 100 kPa to 1000 kPa (one point per 100kPa) and a temperature range from −45 °C to 65 °C (one point per 10 °C or 20 °C). Figure 7b shows differential pressure sensitivities of the resonators, which were quantified as 11.89 Hz/kPa with correlation coefficient of 0.999993. Figure 7c shows the plot of the differential pressure sensitivities of four sensors, indicating a nonlinearity of 0.01% within the pressure range of 100 kPa to 1000 kPa. By comparing the pressure sensors with glass or SOG caps, it was found that the temperature sensitivities of the sensors decreased from 18.32 Hz/°C to ~1 Hz/°C, as shown in Figure 7d. Polynomial surface fitting with the calibration data was conducted to obtain the difference (error) between the experimental results and the compensated pressure value [18]. Figure 7e shows the fitting result of the proposed resonant pressure micro sensor in the full pressure scale (100 kPa~1000 kPa) and temperature range (−45 °C to 65 °C). It was observed that the errors were within ±90 Pa with an accuracy better than ±0.01% F.S (1000kPa). As a consequence, it could be concluded that temperature-insensitive resonant pressure sensor could be realized by the SOG vacuum packaging. 

## 5. Conclusions

A temperature-insensitive resonant pressure sensor based on the SOG vacuum packaging is presented in this paper. The SOG cap not only provides hermetic sealing for the resonators but offsets the residual thermal stress between the SOI wafer and the glass cap. The experimental results shows the quality factors of the resonators were higher than 16,000 with a differential pressure sensitivity of 11.89 Hz/kPa. Further characterizations based on a closed-loop self-oscillating system indicate that the proposed sensor feature with low nonlinearity within 0.01% F.S and low fitting errors within 0.01% F.S under the pressure range of 100 kPa to 1000 kPa in temperature range of −45 °C to 65 °C. In addition, with the SOG cap, the temperature sensitivity of the sensor dropped from 18.32 Hz/°C to ~1 Hz/°C. The effective structure could be further developed to improve the performance of the resonant pressure sensor.

## Figures and Tables

**Figure 1 sensors-19-03866-f001:**
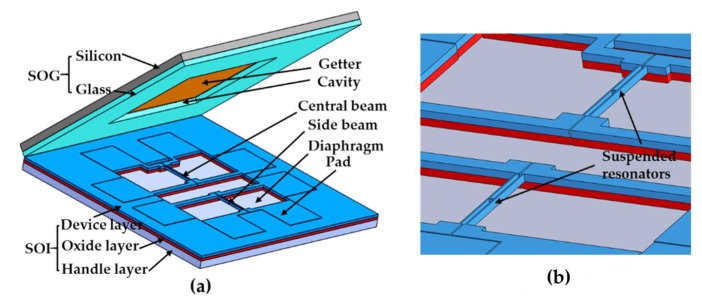
(**a**) Schematic of the resonant pressure sensor, including an silicon-on-insulator (SOI) wafer and an silicon-on-glass (SOG) cap. In the SOI wafer, there is a square pressure sensitive diaphragm in the handle layer, two H-shaped doubly-clamped resonators in the device layer. The SOG cap with a cavity, sputtered with the getter material, is used to form a vacuum packaging for the resonators and reduce the temperature sensitivity of the resonant pressure sensor; (**b**) Zoom in for the key portion of Figure 1a; (**c**) Pressure under measurements causes the deformation of the pressure-sensitive diaphragm, leading to frequency shift of the resonators; (**d**) Electromagnetic excitation/electromagnetic detection, the resonator experiences a Lorentz force to excite the beam into vibration, and the resulting vibrations are sensed by detecting the magnetically induced voltage, developed across the beam.

**Figure 2 sensors-19-03866-f002:**
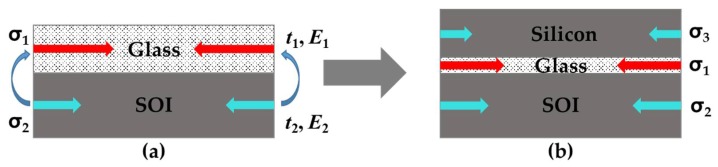
Schematic of multi-layer structures: the SOI wafer bonded with (**a**) the pure glass cap and (**b**) the SOG cap.

**Figure 3 sensors-19-03866-f003:**
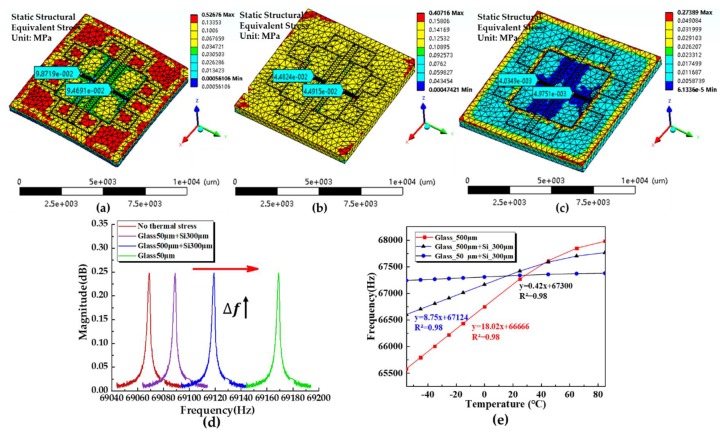
Simulation results of finite element analysis based on ANSYS: (**a**–**c**) Distributions of residual thermal stresses for the sensor chips capped with the pure glass cap (500 μm in thickness), the SOG cap I (300 μm-thick silicon and 500 μm-thick glass) and the SOG cap II (300 μm-thick silicon and 50 μm-thick glass) from left to right; (**d**) Intrinsic frequency shifts of three types of senor chips in response to applied pressure; (**e**) Intrinsic frequency shifts of three types of senor chips in response to temperature variation.

**Figure 4 sensors-19-03866-f004:**
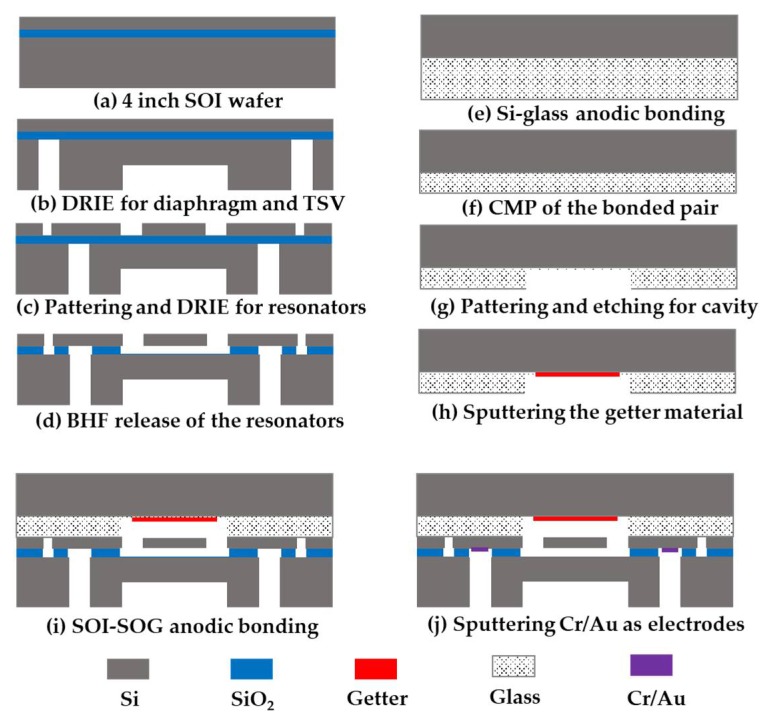
Fabrication process of the proposed resonant pressure micro sensor: (**a**–**d**) Fabrication of the SOI wafer; (**e**–**h**) Fabrication of the SOG cap; (**i**) SOI-SOG anodic bonding; (**j**) Fabrication of electrodes.

**Figure 5 sensors-19-03866-f005:**
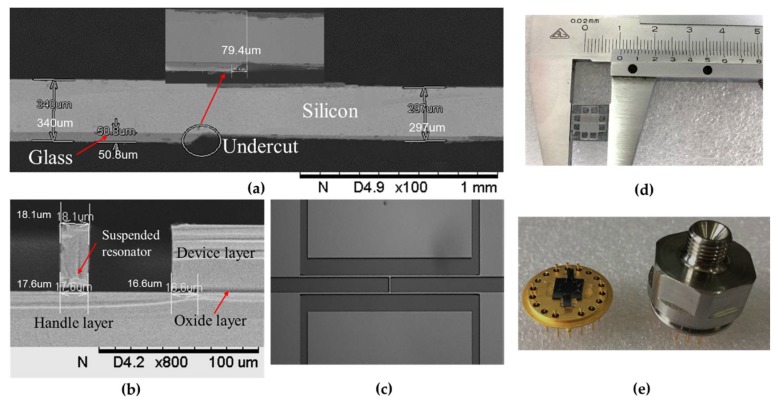
(**a**) The scanning electron microscopy (SEM) image of the cross section of the SOG cap; (**b**) Cross section of the suspended resonator with an undercut of 16.6 μm on the anchor; (**c**) Image of the top view of the resonator; (**d**) Image of a prototype sensor with a dimension of 10 mm × 10 mm; (**e**) Image of packaged prototype of the resonant pressure micro sensor.

**Figure 6 sensors-19-03866-f006:**
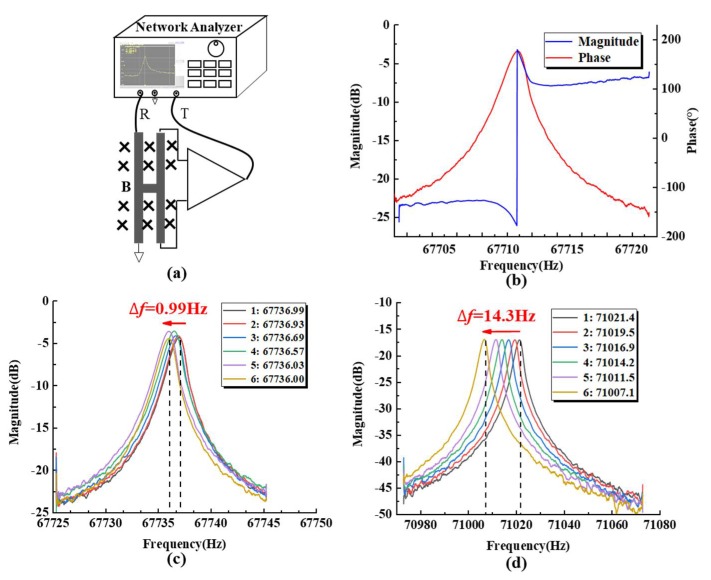
(**a**) Schematic of the open-loop platform for the resonators; (**b**) open loop result of the sensor, recording a beam resonant frequency of ~67736Hz (quality factor of 16589); multiple cycles of frequency responses of resonators based on SOG, (**c**) glass, and (**d**) vacuum packing.

**Figure 7 sensors-19-03866-f007:**
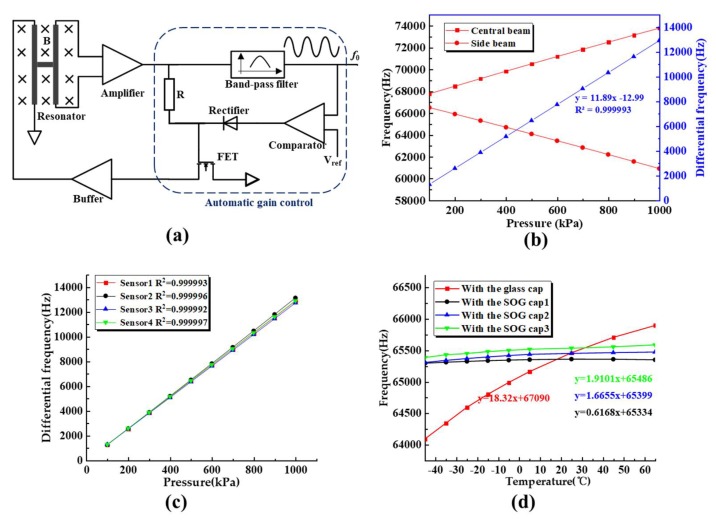
Characterization results: (**a**) Schematic of the self-oscillation system including an amplifier module, a drive buffer module and an automatic gain control (AGC) circuit; (**b**) Pressure sensitivities of the sensors with the pressure varying from 100 kPa to 1000 kPa at room temperature; (**c**) Differential pressure sensitivities of four pressure sensors, indicating a nonlinearity of 0.01% within the pressure range of 100 kPa to 1000 kPa; (**d**) Temperature sensitivity of the pressure sensor with different caps under an applied pressure of 100kPa; (**e**) Fitting errors of the proposed sensor within the full pressure range of 100 kPa to 1000 kPa and temperature range of −45 °C to 65 °C.

**Table 1 sensors-19-03866-t001:** Properties of materials used in finite element method.

Items		Silicon	BF33
Young’s modulus (GPa)		165	64
Density (g/cm3)		2.33	2.23
Poisson’s ratio		0.28	0.2
CTE (ppm/°C)	20 °C	2.46	3.25
	250 °C	3.61	3.25
	500 °C	4.15	3.5

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
