# Peer review of "A Temperature-Insensitive Resonant Pressure Micro Sensor Based on Silicon-on-Glass Vacuum Packaging"

_sensors, 2019, doi:10.3390/s19183866_

Round 1

Reviewer 1 Report

This work report on the development of resonant pressure sensors using silicon-on-glass bonding. The sensors operate based on the shifting of resonant frequency due to the change of pressure caused by a deflection in a silicon diaphragm. One of the main issues in pressure sensing is temperature, which adversely affects sensor output. Therefore, the design and development of sensors that are less temperature-dependent is of significant interest. By employing the silicon on glass as a sealing layer for pressure sensors, the authors markedly reduced the thermal stress; therefore, minimized the residual thermal stress in micro-resonators. The reviewer recommends this work for publication in Sensors after a major revision. Please address the following comments:

The maximum residual thermal stress in the simulation is relatively large (approximately 0.5 for pure glass; 0.4 for SOG cap I and 0.3 for SOG cap II), Figure 2(A). This large strain may exceed the tensile strength of silicon materials. Please comment on this point. Please add more qualitative discuss on why a combination of silicon and glass works better than glass capped structure only. For instance, how can the stress be balanced using bi-layer instead of a single layer (qualitatively) Does the ratio between the thickness of Si and glass affect the thermal stress? What is the role of the getter layer? Explanation of the getter layer is not clear in this manuscript. Did the author conduct the simulation with and without the getter layer? In fact, the sensors were released from the silicon substrate (after etching the SiO2 sacrificial layer.) However, figure 1(a) does not clearly show that the resonator was released. The reviewer suggests the authors to enhance the sketch of Figure 1(a) and (b) so that the readers will have a better view (for instance adding a zoom in or inset figure showing that only the resonator parts were released.) In figure 4, please provide an SEM image of suspended resonators. How did the authors measure the resonant frequency of the resonators? Did the author use a laser Doppler system to measure the natural resonant frequency caused by the Brownian motion? If so, the requirement of the optical setup would make the system bulky which is not suitable in practical applications. In figure 5(c), why did the frequency shift after multiple cycle testing? The frequency is expected to be the same if a similar pressure level is maintained. Most of the references in the manuscript were published more than 4 years ago, and there is no paper on a similar topic published recently. The reviewer suggests the authors to cite more recent papers on this topic. For instance, the anodic bonding was used in this process, and the glass substrate can work at a relatively high temperature. This could be considered an advantage of the proposed structure. Please cite the following paper on anodic bonding: DOI: 10.1109/LED.2018.2808329; 10.1002/admi.201800764; 10.1021/acsami.7b06661 There are some typos in the manuscript. Please revise. For instance:

Line 27: Compared to; not compared with

Line 140: Figure 4(c); not Figure 5(c)

Reviewer 2 Report

Several typos are significantly affecting the experimental description

Regardless typos, manuscript requires grammar revision

References that are more recent are missing

Before publication, this paper needs additional analysis for the following technical items:

          Regarding one of the main contributions for this paper, authors are arguing the following: “The SOG cap not only provides hermetic sealing for the resonators but offsets the residual thermal stress between the SOI wafer and the glass cap.” This asseveration is supported by experimental results and some numerical simulations, but the thermomechanical mechanism must be analyzed and explained.

          If we consider automotive applications, the temperature analysis (-45C to 65C, Fig 6) is reduced towards needed higher temperature, additional comments are needed.

          Which type of applications are considered for this sensor?

          Regarding Fig 3d and related text, this figure has been reported by the authors (https://link.springer.com/referenceworkentry/10.1007%2F978-981-10-2798-7_15-1), however as is evidenced while the beams are released possible and significate lateral etching could occur into the oxide film supporting the rest of the structure, hence additional description is needed.

Round 2

Reviewer 1 Report

The authors have address all of my comments. Therefore, I would like to recommend this work for publication in Sensors as is.